# The More, the Better? Improving VR Firefighting Training System with Realistic Firefighter Tools as Controllers

**DOI:** 10.3390/s21217193

**Published:** 2021-10-29

**Authors:** Seunggon Jeon, Seungwon Paik, Ungyeon Yang, Patrick C. Shih, Kyungsik Han

**Affiliations:** 1Department of Artificial Intelligence, Ajou University, Suwon 16499, Korea; vskylife5901@ajou.ac.kr (S.J.); swp0514@ajou.ac.kr (S.P.); 2Electronics and Telecommunications Research Institute, Daejeon 34129, Korea; uyyang@etri.re.kr; 3School of Informatics, Computing, and Engineering, Indiana University Bloomington, Bloomington, IL 47405, USA; patshih@indiana.edu; 4Department of Intelligence Computing, Hanyang University, Seoul 04763, Korea

**Keywords:** virtual reality, firefighting training system, VR control modality, user experience, presence, cognitive load

## Abstract

A virtual reality (VR) controller plays a key role in supporting interactions between users and the virtual environment. This paper investigates the relationship between the user experience and VR control device modality. We developed a VR firefighting training system integrated with four control devices adapted from real firefighting tools. We iteratively improved the controllers and VR system through a pilot study with six participants and conducted a user study with 30 participants to assess two salient human factor constructs—*perceived presence* and *cognitive load*—with three device modality conditions (two standard VR controllers, four real tools, and a hybrid of one real tool and one standard VR controller). We found that having more realistic devices that simulate real tools does not necessarily guarantee a higher level of user experience, highlighting a strategic approach to the development and utilization of VR control devices. Our study gives empirical insights on establishing appropriate combinations of VR control device modality in the context of field-based VR simulation and training.

## 1. Introduction

Virtual reality (VR) has gained much attention as a technology that is suitable for training services in many domains because it provides trainees with realistic experiences and methods for coping with various situations [1,2,3,4,5,6]. VR has not only significantly lowered cost and space requirements compared to a real training environment but has also allowed trainees to conduct repetitive training at any time without material and environmental restrictions, further increasing its impact, educational value, and a trainee’s self-efficacy.

Providing realistic VR training scenarios is of the utmost importance in order to obtain maximum task and knowledge transfer to real-world settings. A standard VR controller is a key device for supporting interactions between a player and the VR content. It provides key functions, such as triggering, touching, and gripping. However, in a VR training program that includes the use of tools that have different form factors and provide different functions, relying only on the standard VR controller may have several limitations as follows. First, it may not fully reflect the reality of training. For example, firefighters use many tools (e.g., fire hoses, fire axes) in training and in the field. However, since these tools are different from the standard VR controller in terms of shape and size, it can be difficult for users to learn the exact usage of the tools through the standard VR controller. Second, the sense of realism in the training can be low when using the standard VR controller; even if visual representation of the controller in the VR environment is the same as the real tool, the user may experience cognitive inconsistency due to tactile differences. Finally, there often exist situations in which multiple tools must be used in combination [2,7]. For example, to rescue a person, firefighters may have to use a flashlight to find their way in the smoke or use a fire axe to remove an obstacle in their way. Because it is difficult to perform complex actions with only the standard VR controller, such a VR environment may not be able to properly foster and sustain the sense of presence that is experienced at an actual fire site.

To address these problems, the development of real tools as auxiliary or even primary VR controllers should be considered. In many training situations, a variety of tools are required for rescue and evacuation. Much effort has been made to develop more realistic, physical VR controllers that give users greater flexibility and control in the VR environment [8,9,10,11,12,13]. However, while these technological advances to increase realism are critical, it should be noted that they do not always guarantee a high level of user experience.

User experience is affected by a variety of human factor constructs. One of the key human factor constructs, considered in human–computer interaction, is *cognitive workload* when it comes to the use of tools or computer systems. Cognitive workload is defined as the interaction between the demands of a task that an individual experiences and his/her ability to cope with these demands [14,15,16]. In the VR context, this means that although having more VR devices could lead to a greater sense of realism, one’s working memory resources could be overloaded when extraneous activities of using/controlling multiple tools are processed. Therefore, understanding the association between VR controllers and user experience helps to establish the direction of effective VR content/system development by detailing the relative strengths and weaknesses of real tools and the standard VR controller in terms of high sensibility and cognitive burden in the context of training.

Thus far, very few studies have examined the relationship between the new tools and human factors, such as how the affordances of a newly developed tool positively or negatively impact user experiences in VR training compared to the standard VR controller, and how the design of VR controllers could be leveraged to improve the user experience in VR training. Therefore, the research objective of this paper is to articulate the relationship between the use of VR controllers (i.e., standard VR controllers and customized controllers that mirror real tools) and the user experience through the lenses of two human factor constructs—*perceived presence* and *cognitive load*. Our research context is firefighting training to which VR technology has started to be applied [2,17,18,19,20]. Through interviews with firefighters, we identified four core firefighting tools used in training and in the field. We then developed VR controllers that support the same operations as the real ones. With the tools that we iteratively designed and prototyped with the support of our local fire department, we first conducted a pilot study with six participants to test the technical aspects (e.g., calibration, functionalities) of the tools, VR locomotion, and training scenarios. After addressing issues of the VR system identified in the pilot study, we conducted a user study with 30 participants.

Our user study introduced a hybrid method that uses a real tool together with a standard VR controller. To achieve our research goals, we investigated the effectiveness of the hybrid method while controlling for the number of control devices (i.e., isolating the factor that might influence the results of the dependent variables). The study results indicated that the hybrid group had generally higher perceived presence and lower cognitive load than the control and real tool groups. These results (1) demonstrate that increasing realism by having more realistic devices does not always lead to greater user experience and (2) highlight the strategic use of real tools in combination with standard VR controllers to optimize user experience and achieve training goals.

The contributions of this research are as follows.

We developed a firefighting training system and integrated real firefighting tools and the standard VR controller to provide users with an immersive VR experience for training.We analyzed the relationship between the VR controller and human factor constructs (i.e., perceived presence and cognitive load) at different levels of tool modality (i.e., standard VR controllers only, real tools only, and hybrid).We present a strategic plan for the use of VR controllers to help enhance the user experience and achieve VR training goals.

## 2. Related Work

### 2.1. VR Simulation for Training

The use of VR technology allows training for incidents that cannot easily be replicated and are costly to recreate. It is cost-effective, supports complex and varied training scenarios, offers increased safety for high-risk training, and allows data recording [5]. This applies to many cases of serious training, such as training for firefighting. It is possible for firefighters to perform practical firefighting missions in an environment that is realized like a real fire situation, and to prepare for emergencies and unpredictable situations in advance. Engelbrecht et al. [5] highlighted the strengths (e.g., cost effective, complex and varied scenarios, data recording) and opportunities (e.g., increase in physical fidelity, increased resilience against adverse effects) of VR firefighting training. While live fire training has been the gold standard of replicating the perilous situations firefighters encounter on response calls, many firefighting agencies have started to consider adopting VR technology. For example, the US Fire Administration (USFD) advocates the use of VR training exercises (https://bit.ly/3jKcThe (accessed on 29 October 2021)), and Cosumnes Fire Department in California, USA, conducted firefighter training in VR. Fire trainees were trained on how to deal with dangerous emergencies with VR and how to use new tools and technologies in VR-controlled environments. Australia’s FLAIM Systems develops VR training simulators for firefighters, enabling firefighters to immerse themselves in virtual scenarios that simulate dangerous situations (https://cnn.it/3r6QXxN (accessed on 29 October 2021)).

A growing body of research on VR and firefighting training has been conducted and proved the effectiveness of VR training. Jung et al. [17] presented a VR training environment that uses multisensors, such as vision, audio, tactile, and odor and demonstrated the increased confidence that the trainees experienced after training. Cakiroglu and Gokoglu [18] presented a VR training system where a scenario and mission is adjusted to the behaviors of a trainee and a level of fire and smoke. Clifford et al. [19] developed a VR-based air attack supervisor (AAS) program for firefighters that suppresses large wildfires using helicopters. Jeon et al. [2] identified key requirements that need to be implemented in VR firefighting training and developed a training system that incorporates such requirements providing increased presence and immersive experience. Lovreglio et al. [6] demonstrated that a VR system supporting training for the use of a fire extinguisher was more effective than training through a video with respect to knowledge acquisition, retention of information, and self-efficacy. Bliss et al. [21] considered three environments of firefighting training (i.e., blueprint, VR, and no pre-training) and showed that learning about fire situations in advance through the VR training environment was helpful in performing a given task in actual onsite training.

### 2.2. Customized VR Controller Development

Many previous studies have tried to provide users with a higher sense of reality and bodily experience [7] by matching the haptic provided by the standard VR controller with the visual appearance of the controller inside the VR content. For example, Arora et al. [9] attempted to solve the limitation of the existing controller—that the shape is fixed and cannot be changed—which deteriorates the physical reality. The authors presented VirtualBricks, a LEGO-based toolkit that enables the construction of a variety of physical-manipulation-enabled controllers for VR. Zenner and Krüger [8] introduced a fan-looking controller that allows users to feel air resistance by adjusting the shape of the tool according to the degree of air pressure. Zhu et al. [10] presented Rubik’s Twist, a type of low-cost twistable artifact, to create haptic proxies for various hand-graspable VR objects. Krekhov et al. [22] presented a self-transforming controller that adapts to the current virtual weapon by transforming between a pistol-like controller and a two-handed rifle-like device. In the context of firefighting training, Nahavandi et al. [20] introduced a physical fire hose that can be used in the VR training by modeling physics of water sprayed from the fire hose and developing the point of fire, the rate of spread, and structural changes affected by the fire. These studies all confirmed that users have the most natural experience when there is consistency between the appearance of the controller shown in the VR content and that of the actual controller. This is also in part highlighted in other studies involving training with a limited condition [2,23] (e.g., use of a standard VR controller, participation while sitting on a chair).

However, those studies of customized VR controllers focused on the use of a single controller, and many aspects of the user experience of multiple VR tools remain unexplored. This is important because while having more VR devices could lead to a greater sense of realism, one’s working memory resources could be overloaded when extraneous activities of using/controlling multiple tools are processed, which could deteriorate the effectiveness of training. To the best of our knowledge, no research has investigated the influence of VR control device modality on user experience in training. Therefore, in this work, we developed multiple firefighting tools that are considered important in actual firefighting training and integrated them into the VR firefighting training system. With our VR system, we in particular aimed to investigate associations between the level of tool modality (i.e., four real firefighting tools and the standard VR controller) and two human factor constructs: perceived presence and cognitive load.

### 2.3. User Experience in VR

#### 2.3.1. Perceived Presence

A user’s perceived presence is an important psychological construct that determines the level of user experience of the system, which is defined in various ways. Rizzo et al. [24] defined perceived presence as “the experience a person has when in a VE of being there”. Similarly, Heeter [25] stated a “subjective personal presence is a measure of the extent to which and the reasons why you feel like you are in a virtual world”. Many theories on perceived presence highlight the subjective sense of “being there”, which is experienced during immersion in a virtual environment [26]. According to Schubert et al. [27], perceived presence involves multiple cognitive processes consisting of the sense of being there, which is referred to as spatial presence, attention to real and virtual environments, which is referred to as involvement, and the degree of reflection of reality, which is referred to as realness. In this study, we used the definition of perceived presence by Schubert et al. [27].

#### 2.3.2. Cognitive Load

Emergency training is characterized by a mixture of factors, such as training content, training environment, use of tools, and decision-making, which increase the complexity of the training and entail many cognitive processes. This also applies to emergency training in the virtual environment. Cognitive load refers to the amount of information that the working memory can hold at one time. It is the individual’s cognitive capacity for learning a task, solving a problem, etc. There are two approaches [14,16] to operationalize cognitive load. The primary idea of both is that, if the cognitive load exceeds an individual’s processing capacity, s/he will struggle to successfully complete the task.

The first approach is based on three dimensions: intrinsic cognitive load, extraneous cognitive load, and working memory [16]. Intrinsic cognitive load is concerned with the intrinsic complexity of the information or material itself, extraneous cognitive load is concerned with how instructions are designed, and germane memory concerns learner characteristics. Related studies have shown the effectiveness of instructional strategies that lower cognitive load by reducing task complexity (intrinsic cognitive load), such as isolation and integrated presentation of information [28], or a modular presentation of complex information [29].

The second approach involves two dimensions: mental load and mental effort [14]. Mental load pertains to the capacity to deal with a task, and mental effort relates to an individual’s cognitive capacity for task execution and management. Much research has investigated the relationship between cognitive loads and learning outcomes. In one study, participants showed high mental load and mental effort for a task with high complexity [14]. Other studies found that higher mental load leads to lower learning satisfaction and knowledge test scores in the VR environment [30] and that proper use of leaderboards and penalties can increase game efficiency (e.g., retention learning to task, recall process) [31]. Some studies have looked into using machine learning techniques to learn one’s level of mental load based on physiological signals, such as electrodermal activity and ocular activity [32,33].

In this work, we employed the second approach, considering mental load and mental effort, because these deal with a more general aspect of an individual’s cognitive capacity toward a task and provide an educational assessment that can be used to better understand task difficulty and task performance.

## 3. Study Procedure

Before developing the VR firefighting training program, we conducted interviews with five firefighters to understand the background of firefighting training and the demand for a firefighting training program [2]. Three firefighters were from a local fire station, and two firefighters were from a fire training school. In the interviews, we focused on identifying factors (e.g., self-control in movement, disturbance of view and dark environment, easy manipulation) that need to be considered when creating a VR training environment for a realistic fire situation. Firefighters shared that an underground establishment is quite dangerous because it is an isolated and narrow space, so we used it as the location in our VR scenario and the content of our VR system. Our study was divided into three phases (Figure 1).

Investigation: Through the interviews with firefighters, we identified basic yet important firefighting tools and factors that need to be considered to support realistic experiences in VR firefighting training.

Development: We designed and developed four real firefighting tools (i.e., fire hose, fire axe, flashlight, and air pressure gauge) and integrated them into the VR firefighting training system. We also developed methods to track hand control and movement and applied them to the training system. Specifically, we designed the VR system that accommodates different levels of VR device modality from the standard VR controller and real firefighting tools.

User study: We conducted two user studies. One was a pilot study designed to examine the use of the four tools in the VR scenario and improve their usability. The other study was a main user study aiming to investigate associations between the level of VR device modality (i.e., two standard VR controllers, four real tools, and a hybrid of a standard VR controller and a real tool) and two salient human factor constructs (i.e., perceived presence and cognitive load). In the main study, we aimed to answer two hypotheses that pertain to the relationship between them as follows.

H1: Using one real tool and one standard VR controller will result in a similar degree of perceived presence as using four real tools.H2: Using one real tool and one standard VR controller will result in a similar degree of cognitive load as using two standard VR controllers.

Based on the study results, we discuss answers to the hypotheses and present actionable insights on effective ways to use real tools together with standard VR controllers to improve user experience in VR training systems.

## 4. Development

### 4.1. VR Environment and Scenario

To support a high degree of freedom in the training, our VR system allows a player to freely move around and use firefighting tools. At the start of the program, a player receives a briefing from a senior firefighter about the situation inside the VR and about the mission, which is to rescue an isolated person inside a room located on the basement level of a commercial establishment. Upon completion of the briefing, the player picks up an oxygen mask, puts it on, and enters the basement entrance. There are many small rooms on the basement level, and the player checks through rooms to locate the fallen victim at their own discretion. During this process, several dangerous events (e.g., electrical short circuit, flashover caused by thermal radiation feedback, backdraft caused by a smoke explosion, broken windows) may occur randomly. When the player finds the isolated person, the player changes to a state supporting the rescued person, and the training scenario ends when they escape through the building entrance. Figure 2 (left) illustrates a VR scenario environment considered in this study.

### 4.2. Firefighting Tools

We investigated the types of tools that are most frequently used during firefighting through the interviews with the firefighters. Four core firefighting tools were identified, and each tool is illustrated in Figure 3. This section explains the development process of these firefighting tools.

#### 4.2.1. Fire Hose

A fire hose is one of the firefighter’s most important tools. It is largely composed of a hose through which water passes and a fire nozzle through which water is sprayed (Figure 3a). The pipe window is composed of a fixed part connected to the hose and a rotating part that can be rotated by the firefighter, as shown in Figure 4a. The rotating part is normally locked to the maximum setting; when a firefighter turns it to the unlock setting, water begins to come out of the nozzle. The greater the degree of rotation, the greater the spray angle of the water. By adjusting the rotation, the firefighter can adjust the response based on the size of the fire.

To make a fire hose into a VR controller (Figure 4a), we attached a sensor to the pipe window part of the fire hose. We used a rotary encoder (Arduino KY-040 Rotary encoder) to measure the degree to which the player turns the rotating part. The value measured by the sensor was sent to a Unity engine (and our VR system) using Arduino serial communication of Arduino Uno (https://store.arduino.cc/usa/arduino-uno-rev3 (accessed on 29 October 2021)). The angle and distance of the water sprayed from the player’s hose can be adjusted based on the received result. We performed 3D modeling to represent the shape of the fire hose inside the VR, and Obi-rope [13] was used to represent the curve of the hose.

#### 4.2.2. Fire Axe

Another important firefighting tool is a fire axe, which is primarily used to create paths by clearing obstacles (e.g., windows, doors, items) around the player. As illustrated in Figure 3b, there is a part of the axe that can be inserted into the doorknob. If it is rotated while applying force, the doorknob is easily separated from the door according to the lever principle.

We made an object in the shape of the real axe using a 3D printer with a wood pole. We attached a VIVE tracker to the axe, tracking the location of the axe and making the axe interact/communicate with objects (e.g., door knob) in the VR content. All of these operations were implemented with the physics engine of Unity. We applied the operation processes of the fire axe to the VR system. For example, when the player puts the door opener part of the axe on the doorknob inside the VR and rotates it at a high speed, the system detects the speed and separates the doorknob from the door. After that, the door with the broken handle is converted to an open state, and the player can enter the room. If the player hits a window with the axe, the window will break.

#### 4.2.3. Flashlight

In fire situations, a firefighter’s visibility could be completely blocked due to very heavy smoke. In such cases, firefighters can barely even see their hands. This is why firefighters use flashlights when navigating dark spaces (Figure 3c).

To operate the flashlight inside the VR, we designed the system to detect whether the flashlight button was pressed by connecting the power switch part of an actual flashlight to Arduino. When the player presses the button, it is transmitted to Unity using Bluetooth communication, and the power state of the flashlight inside the VR changes (Figure 4b). When the light is turned on, we programmed the system to reduce the amount of smoke particles when colliding with the light so that the player can navigate inside the space with relatively clear sight.

#### 4.2.4. Air Pressure Gauge

In actual fire situations, suffocation from smoke is one of the leading causes of death. Thus, the air tank is important, and our VR system allows the player to check the oxygen level gauge connected to the air tank (Figure 3d). The air pressure gauge usually starts from the point marked 300 and drops to 0. When the remaining oxygen level falls to the red zone, a loud whistling sound is generated, indicating the low level of remaining oxygen to the player and asking the player to take appropriate action.

In general, an actual air tank can be used for about 30 min at a time; however, considering possible VR-related side effects, such as motion sickness and headache, we set the time of the VR training to a maximum of 15 min. The hand of the air pressure gauge gradually rotates over time so that the player can easily check how much time is left. As with the development of the fire axe, it was not cost-effective to use a real air tank. Thus we made the air pressure gauge with 3D printing. The firefighters we interviewed mentioned that using the air pressure gauge would be sufficient for the purpose of training.

### 4.3. Movement

#### 4.3.1. Hand Control

To enable natural interactions between the player and the VR content, our VR system supports basic interactions, such as grabbing surrounding objects and interacting with collisions, in addition to the interactions with the firefighting tools. To do this, we used the Leap motion sensor (https://www.ultraleap.com/ (accessed on 29 October 2021)). After attaching the Leap motion sensor to the front of the head mount display (HMD), we were able to make the VR system recognize the player’s hand and implement interactions (e.g., opening a door, picking up objects). In this way, we tried to provide a realistic experience to the player in VR.

#### 4.3.2. Locomotion

In many VR training programs, players have to use a standard VR controller to move. However, using a standard VR controller only for locomotion limits other interactions. To mitigate this, we configured the environment to support the player’s movement in VR using walk-in-place (WIP). After placing a VIVE tracker on the player’s waist and both feet, our algorithm determined whether to advance the player’s position based on the coordinates and direction of the tracker. To implement WIP, we attached the HTC VIVE tracker to plain shoes and detected the height of the tracker, $h_{stand}$, when the player stood (i.e., the feet were on the floor). When the player lifted one foot, we measured the maximum height of the foot, $h_{foot}$. We also detected the height of the HMD, $h_{HMD}$. We then set the threshold of the player’s movement based on the change between $h_{stand}$ and $h_{foot}$. Through many trials, we determined that the most natural proportion of the threshold of the change is about 15% of $h_{HMD}$. The threshold, *t* was calculated as follows:$$t_{\left(h_{foot}-h_{stand}\right)}>h_{HMD}*0.15$$

As *t* varies by a player, we designed our system to recognize a player’s step. When the change in the height of the foot was greater than *t*, and then went below it within 0.1 s, the system moved the player in the direction of the VIVE tracker attached to the player’s waist. In this way, the player could move freely in VR without a standard VR controller We are aware that research in VR locomotion is very active and our tracking method in the user studies may not be effective to fully support a player’s movement. However, no participants mentioned difficulties in movement during the experiments. Thus our method was sufficient enough to address our study goals.

### 4.4. Integration

After the development of the VR scenario, real firefighter tools, and movement were completed, we integrated them to create the VR training system. Figure 5 shows a player wearing and carrying firefighting tools. The flashlight and the air pressure gauge were connected to the vest so that the player could carry other tools in his/her hands.

## 5. Pilot Study: Validation of VR Controller

### 5.1. Study Procedure

The experiment was conducted in a university laboratory with a total of six participants (university students; mean age: 26.3, SD: 3.2) who were recruited through a university bulletin board or word-of-mouth. We used the HTC VIVE PRO HMD, and the study was conducted based on Unity 3D in a Windows 10 system equipped with Intel Core i7, RAM 16GB, and GeForce GTX 1070.

The study procedure was as follows. The participants were given verbal explanations on how to use the firefighting tools. They went through a five-minute tutorial to experience the use of tools and the VR system (Figure 6 illustrates some examples of the tutorial). They wore VIVE-tracker-fitted shoes for locomotion. We then asked the participants to play in the VR scenario and follow the instructions on the real tool use when the instruction prompt appeared. After the trial, the participants were asked to provide feedback on topics, including the strengths and weaknesses of using the tools and suggestions for improvement through the interview. The study took about 25 min on average. Upon completion of the study, the participants were compensated $10 for their time.

All subjects gave their informed consent for inclusion before they participated in the study. The study was conducted in accordance with the Declaration of Helsinki, and the protocol was approved by the Ethics Committee of the internal Institutional Review Board at the authors’ institution (202010-HS-003).

### 5.2. Results

Overall, the participants provided positive feedback on the usability of the four real tools. They mentioned that functions of each tool worked well and that they did not find any specific difficulties in hand control and movement. The participants generally agreed that they were able to operate and interact with the tool while participating in the training. They also mentioned that the length of the scenario was reasonable and were able to experience various firefighting situations. From these responses, we confirmed that the tools were reasonably well designed and operated.

Participants mentioned that they experienced high cognitive load when using multiple real tools because they needed to learn how to use, transport, and replace various tools. Three participants suggested that it might be beneficial to allow users to focus on using one real tool and use the standard VR controller in place of the other three firefighting tools (e.g., *“Some of the tools can be implemented in the standard VR controller”* (P2), *“I am not sure whether carrying all these tools is actually more effective than a single integrated controller”?* (P6)). This comment was also made by a firefighter we interviewed, who believed that selective use of the standard VR controller and real tools might be better for training purposes. These comments are well in line with the motivation of our research and main user study, which is investigating the relationship between VR controller device modality and user experience.

## 6. User Study: VR Control Device Modality

### 6.1. Background and Hypotheses

The objective of the user study was to investigate the relationship between the VR control device modality and user experience. Based on the pilot study results, we defined three conditions for the user study (Figure 2).

Standard controllers condition (control): two standard VR controllers (one for tool selection and the other for tool operation).Real tools condition (experimental #1): four real firefighting tools.Hybrid condition (experimental #2): one real tool and one standard VR controller.

We specifically considered a *hybrid condition* to isolate the factors that might influence cognitive load. We decided to use the fire hose as a real tool and integrated the rest of the tool functionalities into the standard VR controller. Our rationale behind this decision was that a fire hose requires a person to twist the head of the hose with a joint rotation that cannot be physically represented by the standard VR controller. In contrast, a fire axe (which needs rotation and back and forth movement), a flashlight (which controls light through a button press), and an air pressure gauge (which displays the remaining time until oxygen runs out) are relatively easy to implement with a standard VR controller. By limiting the experimental design to a use of the real fire hose and the standard VR controller and controlling for the number of devices (the same as two standard VR controllers), we were able to examine the effects of making one of the controllers more realistic on user experience.

Since it is possible to realistically learn the functional usage of all four firefighting tools, we expected that this hybrid condition would not significantly reduce the player’s perceived presence compared to using all four firefighting tools at once in the VR environment. In addition, we expected that the level of cognitive load would be maintained if only one real firefighting tool was included in the VR system. Thus, the hybrid condition would result in a cognitive load that was comparable to that of using two standard VR controllers.

In summary, we expected that the results of perceived presence in the hybrid condition would be similar to those of the real tools condition; at the same time, the cognitive load in the hybrid condition would be similar to the control condition (using two standard VR controllers). In other words, we expected that the hybrid condition of using a fire hose tool in conjunction with the standard VR controller, which implements the functionalities of fire axe, flash light, and an air pressure gauge, would have the advantages of both approaches. As such, we constructed the two hypotheses (H1 and H2 in Section 3).

### 6.2. User Study Design

We recruited a total of 30 participants (mean age: 24.7, SD: 4.0) through a university bulletin board or word-of-mouth. Twenty-one of them were university students and the rest were not students. The VR scenario and task goal were identical to the pilot study.

We randomly assigned 10 participants to each of the three conditions. We employed a between-subjects experiment to counterbalance across three conditions and to reduce the physical and cognitive burden of going through all three conditions. The execution time for each condition was about 15 min. All conditions ran in the same virtual environment and scenario. After completing the task, participants answered a survey, which consisted of the questions on perceived presence and cognitive load. Lastly, they were asked to have an interview with the researcher.

As for the pilot study, upon completion of the study, the participants were compensated $10 for their time. The study took about 30 min on average.

### 6.3. Measurement of Human Factor Constructs

We utilized the following validated survey instruments to measure perceived presence and cognitive load. A five-point Likert scale was applied to all questions.

#### 6.3.1. Perceived Presence

To measure perceived presence, we used the Igroup Presence Questionnaire (IPQ) [27]. In this survey, presence was composed of four items: general presence (G), spatial presence (SP), involvement (INV), and realness (REAL). This survey has been broadly applied in many VR studies and has been proved to be an appropriate inventory to scale presence [34,35,36,37]. For example, Felnhofer [36] examined differences in social avoidance tendencies and prosocial behaviors during communications with avatars or computer agents. Iachini [37] used IPQ to analyze the relationship between perceived presence and mental imagery.

#### 6.3.2. Cognitive Load

To measure the cognitive load, we used the questionnaire [14,38], which was created from the perspective that cognitive load is composed of mental load and mental effort. It includes six questions for each of mental load and mental effort. Mental load is task-related and indicates the cognitive capacity needed to process the complexity of a task. Mental effort is subject-related and reflects an individual’s invested cognitive capacity while working on a task.

### 6.4. Results

#### 6.4.1. Survey Responses

We measured perceived presence and cognitive load for each group. We used the ANOVA and Tukey posthoc tests for group comparisons. Figure 7a shows the perceived presence scores. For general presence, the ANOVA test showed a marginally significant group difference (*F*(2,27) = 3.16, *p* = 0.055). The posthoc tests showed a marginally significant difference between hybrid and control conditions (*p* = 0.059). For realness, the ANOVA test showed a significant group difference (*F*(2,27) = 4.87, *p* = 0.013), and the posthoc tests showed a significant difference between hybrid and control conditions (*p* = 0.012). Lastly, for the overall perceived presence, the ANOVA test showed statistically a significant group difference (*F*(2,27) = 5.38, *p* = 0.013). The posthoc tests showed significant differences between real tools and control conditions (*p* = 0.042) and hybrid and control conditions (*p* = 0.031). Based on these results, we could conclude that H1 was supported.

Figure 7b shows a comparison of the cognitive load results. For mental load, the ANOVA test showed a significant group difference (*F*(2,27) = 3.67, *p* = 0.032). The posthoc tests showed a significant difference between real tools and hybrid conditions (*p* = 0.043) and a marginally significant difference between real tools and control conditions (*p* = 0.095). For mental effort, the ANOVA test showed a marginally significant group difference (*F*(2,27) = 3.10, *p* = 0.060), and the posthoc tests showed a marginally significant difference between real tools and control condition (*p* = 0.055). For the overall cognitive load, the ANOVA test showed a significant group difference (*F*(2,27) = 5.20, *p* = 0.013), and the posthoc tests showed significant differences between real tools and control conditions (*p* = 0.049) and real tools and hybrid conditions (*p* = 0.012). In summary, the hybrid condition exhibited similarly low cognitive load as the control condition, which is significantly lower than the real tools condition. Thus, H2 was supported.

Based on these results, we found that the hybrid condition incorporated the advantages of using real tools, which improved the sense of presence and the standard VR controllers for convenience and familiarity. The improved sense of presence and less cognitive load in the hybrid condition is more likely to lead to long-term retention and adoption of the VR training system compared to the control condition.

#### 6.4.2. Interviews

We analyzed the user experience feedback by dividing the feedback into two categories as follows:

**Perceived presence and cognitive load in VR training:** As we saw in the survey results, most participants in the control group mentioned that training only with the VR standard controller does not affect VR realism. One participant mentioned, *“Training with the controller is quite easy but I didn’t feel very engaged”* (C-P2, which denotes the second participant in the control condition).

Many participants in the experimental #1 and experimental #2 conditions mentioned that they had a high sense of realism, helping them feel like they were in a real situation. The visual and tactile aspects of the firefighting tools used in the VR supported the participants’ expectations well, meaning that the tools satisfied their mental model. For example, participants mentioned that *“I was quite engaged in the training”* (E1-P1), *“Using the tool and the controller seems quite effective in training engagement”* (E2-P4).

For some participants, especially those who were in the experimental #1 condition, engagement was found to be challenging due to difficulties in the tool replacement process or a situation caused by the twisting of the HMD line and the tools. Some participants mentioned that they had difficulty engaging in the training because of the somewhat bulky fire hose and fire axe; for example, *“I know these tools are essential but still quite difficult to carry”* (E1-P2), *“⋯ personally, I don’t think having more tools always lead to greater realism”* (E1-P5). This finding pertains not only to engagement but also to the cognitive load outcome. The use of four real tools led to a higher cognitive load for various reasons, such as discomfort in motion manipulation (due to the inconvenience in changing firefighting tools) or difficulty in the simultaneous use of the tools.

However, we found that discomfort was not always considered a disadvantage. This reaction was more observed from the participants in the experimental #2 condition than the experimental #1 condition. For example, *“Several trials and errors were required and harder than expected, but needs to be done”* (E1-P7), *“It is somewhat uncomfortable to use the real tools, but the sense of reality was quite high, so I could feel the frustration of the actual firefighters’* (E2-P2), *“Inconvenience represents the actual situation more realistically. When training actual firefighters, a real tool would be more appropriate than the default, standard controller”* (E2-P8).

In summary, participants’ qualitative feedback was quite consistent with the survey results. Both the participants in the experimental #1 condition and those in the experimental #2 condition expressed high realism and engagement in VR training. However, compared to those in the experimental #2 condition, more participants in the experimental #1 condition shared high cognitive load and asked for flexibility of tool choice or use. These results in part highlight the effectiveness of the hybrid approach in training.

**Ways to improve the VR training system and experience:** We specifically asked the participants to share their thoughts on how to use firefighting tools in a more efficient and productive fashion. First, most responses from the control condition relate to having additional control devices for more realistic training. Second, most responses from the two experimental conditions mainly pertain to the flexibility of VR controller; in other words, adapting the use of controllers based on user expertise. Four participants thought that use of the standard VR controller or real tools should be determined based on the user’s familiarity and skill level with the real tools. According to one participant who was in the experimental #2 condition, *“It would be better to use the standard VR controller for new firefighters who are not yet familiar with the fire rescue, and for the experienced trainee, using real tools would be better”* (E2-P9). More experienced firefighters could focus on exploring the VR fire scenario and environment to complete the training because they are already comfortable with using the real tools. However, given that novice firefighters have not yet developed the familiarity and expertise with using real tools, they may experience high cognitive load during VR training because they must navigate an unfamiliar training environment while also learning how to properly use the real tools at the same time. For this reason, participants suggested supporting the use of real tools not at once but allowing the user to choose the tool(s) that s/he wants to learn and giving various simulation environments in which a level of using each tool is different; for example, *“⋯ user could first familiarize with each of the tools independently in various firefighting scenarios”* (E1-P10).

In summary, we believe considering greater flexibility of the use of the standard VR controller and real, customized tools is necessary to improve VR firefighting training and its system. In the next section, we further discuss the implications of our study findings focusing on the design of VR systems.

## 7. Discussion

### 7.1. Summary of the User Studies and Implications

In this paper, we aimed to understand the effects of VR control device modality on two human factor constructs: perceived presence and cognitive load. To do this, we developed a VR system and four essential tools for firefighting training. We conducted a series of two user studies to compare user experiences across three conditions. To the best of our knowledge, our work is the first that investigates the influence of VR control device modality on user experience in (firefighting) training. Below, we present several interesting findings and discussion points.

#### 7.1.1. Designing VR Systems for Beyond Being There

The feeling of “being there” in VR can be improved through various means that increase realism. However, the relationship between realism and one’s VR user experience may not always be positive. VR can increase presence by making the control, scenario, or environment more realistic, but can lower user experience (e.g., increase in cognitive load) if the interaction is overly complex.

This conflict has also appeared in our user study. A higher number of more realistic devices did not lead to greater user experience. Generally, perceived presence was higher for the condition with multiple real tools than the one with the standard VR controllers but was similar to and even less than the hybrid condition. Although having real tools would be very likely to increase perceived presence, there seems to be an upper limit threshold that restricts the amount of realism that is preferred by the users in the VR environment. In other words, if a VR system mimics the real world beyond a certain threshold, user experience could suffer from over-complexity and deviate from the learning/immersive experience that a VR training scenario hopes to provide. Furthermore, carrying real tools and choosing, switching, or manipulating the tools in the real world may not be as difficult or complicated as in the virtual world, because just being in VR could also increase the cognitive burden of the player. These perspectives could explain our finding that the hybrid condition was the most effective one for optimizing perceived presence and cognitive load than other conditions. Instead of aiming to achieve perfect realism, it may be more effective to aim for “beyond being there” by leveraging the strengths of the digital medium to satisfy unmet user needs [39]. Prior research has not sufficiently investigated the role of VR control device modality on user experience; thus, the study results in this paper give additional insights that need to be considered for the design of a VR training system.

#### 7.1.2. Hybrid VR Control Device Modality

Our study results showed that training using only real firefighting tools did not always lead to better user experience outcomes. This result is somewhat similar to prior research where the condition with more sensory feedback does not always positively influence greater user experience [17]. Hence our result suggests that the development of a VR controller requires a strategic approach that considers several perspectives. In addition, the development of real tools requires significant time, cost, and effort. Since the hybrid condition was as effective as the condition that uses multiple real tools, it is more cost-effective to focus on identifying the desired training goal and target the development effort on appropriate combinations of real tools and standard VR controller rather than trying to accommodate all possible real tools, scenarios, and environments.

In our hybrid condition, because the fire hose interactions are difficult to support using the standard VR controller, we kept the real fire hose and used the standard VR controller to simulate the use of fire axe, flashlight, and air pressure gauge. However, this is only one of the various possible hybrid combinations. Depending on the training scenario or goal, we could include one or more of the other real tools. Although the user experience may vary depending on different combinations of tools and standard VR controllers, we believe that our study is meaningful in that it introduced a novel research direction for generating empirical design insights and guidelines by examining the relationship between VR control device modality and human factor considerations. Overall, tools should be selected after careful consideration of various factors, such as the type of VR system to be developed and the target population for training.

### 7.2. Limitations and Future Work

Our study has presented insights and design guidelines for the utilization of VR control devices; however, it has some limitations that should be addressed in future research.

First, although firefighters usually move as a group of two at an actual fire site, our VR system supports single-person interaction. When firefighters act as a group, they carry firefighting tools separately or independently. Thus, it would be necessary to take such characteristics into account in future VR firefighting VR system development efforts and user studies.

Second, although the main target users for our VR system are firefighters, our user studies were conducted mostly with university students. However, our study results are still representative of novice firefighters who have not yet gained sufficient field experience with using firefighting tools. This sentiment was echoed by the firefighters who we interviewed after we showed our VR system to them. However, we think it is necessary to increase the validity and applicability of the research results through future experiments with actual firefighters. In addition, the number of participants may be insufficient, which may reduce the generalizability of the study results. In future user studies, we plan to recruit more participants (both firefighters and individuals with diverse experience in or familiarity with VR) and present comprehensive results. Similar to the study by Bliss et al. [21], we will prepare a training scenario and underlying key interaction components (e.g., control device, sensory effects) that replicate an onsite training environment as the VR content and test whether training in VR will help them complete their mission onsite.

Finally, although non-intuitive interaction mechanisms are partially addressed by applying multiple real firefighting tool manipulation and supporting hand/foot movement in VR, additional intuitive interactions that may influence training experience need to be considered and supported for more realistic experiences and scenarios. For example, the reality and effectiveness of the overall training would be enhanced by updating or adding functions, such as improvement of movement (e.g., omni-directions, running, crawling), more tools (e.g., helmet, gloves), additional sensor stimuli (e.g., heat), communication with the control tower (or training leader), and interaction with various objects in the VR environment (e.g., lifting, moving, or dropping objects). We believe there still exist many more aspects that challenge training in the context of VR, which mostly pertain to the user (e.g., lack of specialization, technology barrier, immaturity of technology, uncertain skill transfer, adverse effects of habituation, adverse effects of engagement stimulation) [5], we will also seek to mitigate these challenges through technical support and design of better user experience support.

## 8. Conclusions

To create more sophisticated and highly learnable VR firefighting training, it is necessary to interact intuitively with the virtual environment. Controllers are among the key VR elements that are explicitly associated with interactions between users and the VR environment and could significantly impact user experience. In this work, we aimed to understand the relationship between the VR control device modality and two human factor constructs: perceived presence and cognitive load. We developed a VR firefighting training system that works with the real firefighting tools and the standard VR controller. By observing the effects of VR control device modality on user experience, we aimed to identify the most suitable combination of controller implementation in supporting realistic and immersive VR training. We first conducted a pilot study to iteratively improve the controllers/VR systems in preparation for the user study. Our user study with a total of 30 participants compared user experience outcomes across three group conditions (two standard VR controllers, four real tools, and a hybrid of one real tool and one standard VR controller). The hybrid condition exhibited a higher level of perceived presence and a lower cognitive load than the real tools condition. These results empirically demonstrated the effectiveness of the hybrid VR control device modality and highlighted that using a higher number of realistic control devices inside the VR environment does not necessarily guarantee better user experiences. The results also suggest a strategic approach to the development of VR tools and training scenarios that could help reduce development cost and overhead. Future VR research should further consider establishing the appropriate VR control device modality for training depending on the context and use.

## Figures and Tables

**Figure 1 sensors-21-07193-f001:**
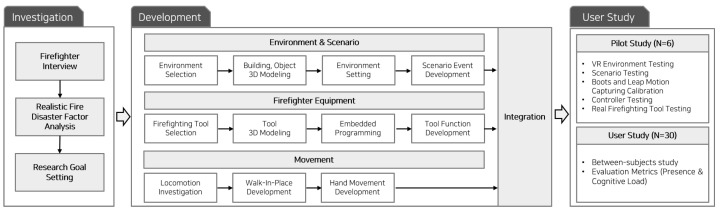
Overall study procedures with three phases: investigation, development, and user study.

**Figure 2 sensors-21-07193-f002:**
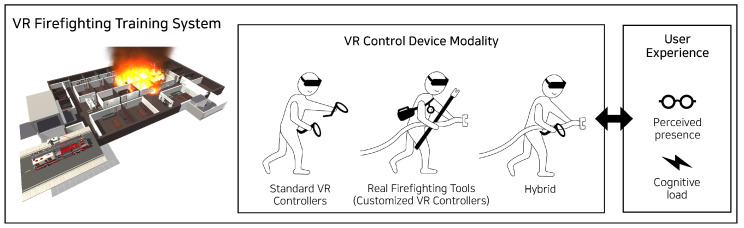
Relationship between the level of VR control device modality and user experience in the VR training context.

**Figure 3 sensors-21-07193-f003:**
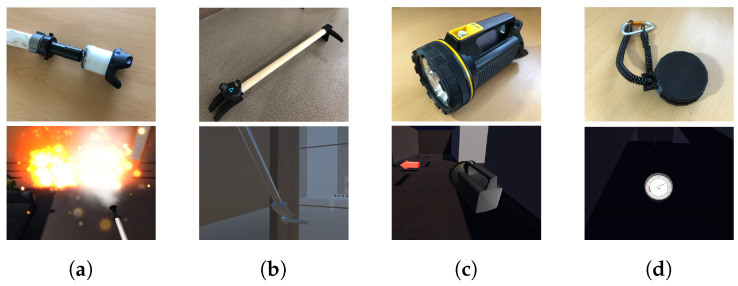
Four real firefighting tools developed in this study: (**a**) fire hose; (**b**) fire axe; (**c**) flashlight; (**d**) air pressure gauge.

**Figure 4 sensors-21-07193-f004:**
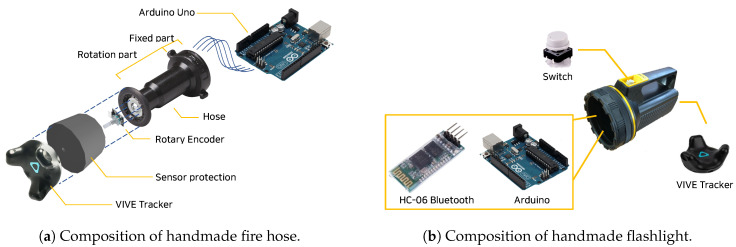
Composition of the (**a**) fire hose and (**b**) flashlight.

**Figure 5 sensors-21-07193-f005:**
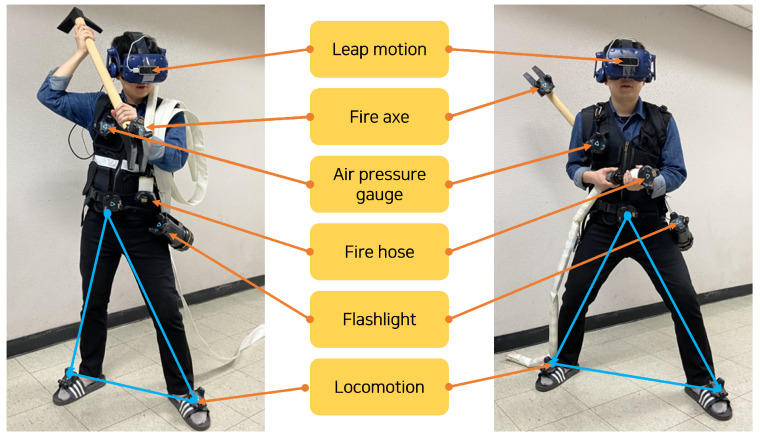
A participant wearing sensor devices and using real firefighting tools for the user study. Three trackers were used to support locomotion (blue lines).

**Figure 6 sensors-21-07193-f006:**
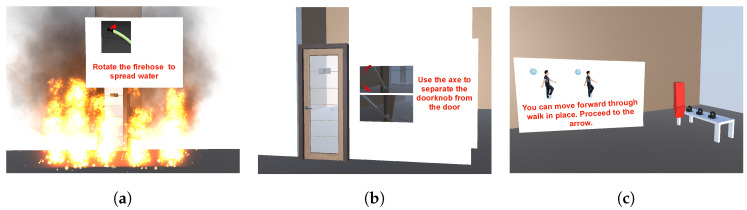
Examples of tutorial scenes before starting the main scenario in the user study. (**a**) Fire hose; (**b**) fire axe; (**c**) locomotion.

**Figure 7 sensors-21-07193-f007:**
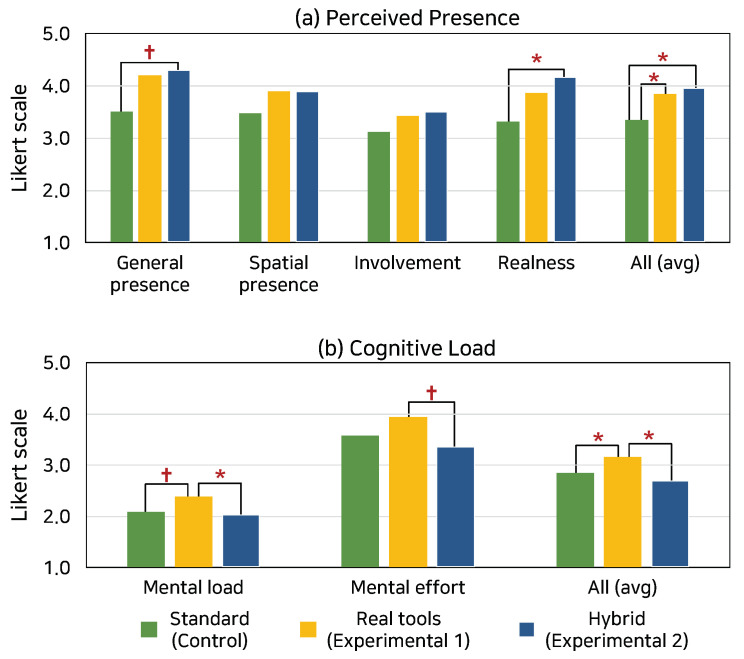
User study results: (**a**) perceived presence and (**b**) cognitive load comparisons across control (two standard VR controllers) condition, experimental #1 (four real tools) condition, and experimental #2 (hybrid with one real tool and one standard VR controller) condition (${}^{†}p<0.10$; ${}^{*}p<0.05$).

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
