# Peer review of "The More, the Better? Improving VR Firefighting Training System with Realistic Firefighter Tools as Controllers"

_sensors, 2021, doi:10.3390/s21217193_

Round 1

Reviewer 1 Report

This manuscript discusses the idea of improving firefighting training systems using VR controllers. The topic is timely and important given the importance of VR manipulation and its effect on improving safety training outcomes. The paper is well-written and well-organized. However, several minor improvements are required. 

  1. The authors need to discuss relevant papers in the background section. Many important references are missing.
  2. The authors need to clearly discuss their results with the past literature in this domain in the discussion section.

Reviewer 2 Report

In this study the authors present a timely and interesting research. The study is well-written, thus the reviewer only has a few remarks.

The reviewer believes that there is little information presented about the fire axe (subsection 4.2.2). How does it send the data back to the computer? With the help of the Leap Motion? If so, it should be elaborated.

Also, the hypotheses are located in the middle of the study. They should be moved towards the beginning of the study, possibly before section 4 to make it more logically constructed.

The language used in the study is fine, only a minor spellcheck is required.

The references are almost in the correct format of the journal, and one important thing should be changed: when citing journal articles, abbreviated journal names should be used instead of full ones.
